# Increased Cardiovascular Mortality in Hemodialysis: The Role of Chronic Inflammation, Complement Activation, and Non-Biocompatibility

**DOI:** 10.3390/toxins17070345

**Published:** 2025-07-10

**Authors:** Ákos Géza Pethő, Tibor Fülöp, Petronella Orosz, Gábor Szénási, Mihály Tapolyai, László Dézsi

**Affiliations:** 1Department of Internal Medicine and Oncology, Faculty of Medicine, Semmelweis University, 1000 Budapest, Hungary; 2Medicine Service, Ralph H. Johnson VA Medical Center, Charleston, SC 29401, USA; tiborfulop.nephro@gmail.com (T.F.); mtapolyai@aol.com (M.T.); 3Division of Nephrology, Department of Medicine, Medical University of South Carolina, Charleston, SC 29401, USA; 4Bethesda Children’s Hospital, 1146 Budapest, Hungary; orosz.petronella@bethesda.hu; 5Institute of Translational Medicine, Semmelweis University, 1085 Budapest, Hungary; szenasi.gabor@semmelweis.hu (G.S.); dezsi.laszlo@semmelweis.hu (L.D.); 6Department of Nephrology, Szent Margit Kórhaz, 1085 Budapest, Hungary

**Keywords:** hemodialysis, peritoneal dialysis, immunological, biocompatibility, mortality, chronic inflammation, complement activation

## Abstract

Background: Chronic kidney disease (CKD) is an established global health problem, with the increased prevalence of vascular inflammation, accelerated atherogenesis, and thrombotic risk all contributing to overall cardiovascular risk. The major CKD-specific risk factor is presumed to be the accumulation of uremic toxins in circulation and tissues, further accelerating the progression of CKD and its co-morbidities, including those of bone mineral disorders and cardiovascular diseases. Materials and Methods: In our narrative review, we focused on non-traditional cardiovascular risk factors, as they evolve with declined kidney function and are potentially further modulated by the choice of kidney replacement therapy. Results: Based on the data from the literature to date, the pre-eminent role of non-traditional risk factors emerges to mediate inflammation and increased cardiovascular mortality. In particular, patients receiving hemodialysis (HD) display dramatically increased CVD-mediated mortality. This intensified state of inflammation may be linked to the direct exposure of the bloodstream to a bio-incompatible environment in HD; for both complement-mediated and non-complement-mediated reactions, the possible contribution of neutrophil extracellular traps and complement activation-related pseudoallergy are reviewed in detail. Conclusions: Our narrative review emphasizes key elements of a bio-incompatible HD environment that may contribute to increased cardiovascular mortality in patients receiving HD. Summarizing these results may provide conceptual opportunities to develop new therapeutic targets.

## 1. Introduction

Chronic kidney disease (CKD) is one of the most widespread non-communicable diseases around the globe, escalating with both industrialization and Westernized lifestyle choices, and it places a significant burden on healthcare systems. Kidney Disease: the Improving Global Outcomes (KDIGO) guidelines classify CKD into five stages based on the glomerular filtration rate (GFR) and three categories according to the level of albuminuria [1]. According to certain studies, it is estimated that over 850 million people are affected by CKD worldwide, a number anticipated to rise as the population ages [2]. This indicates that approximately one in ten adults is affected by CKD [3,4]. CKD is linked to an elevated risk of cardiovascular complications, which can be attributed to the escalating retention of toxic metabolites, electrolyte imbalances, and hypervolemia, among other contributing factors. Not surprisingly, CKD-associated cardiovascular death (CVD) is the leading cause of death in people with CKD [5]. The latest recommendations from the KDIGO indicate that the use of renin–angiotensin system inhibitors and sodium–glucose co-transporter-2 inhibitors (SGLT2 inhibitors) offers significant benefits for patients with CKD. These treatments can help slow the progression of CKD and reduce the risk of renal and cardiovascular events, as well as mortality [1]. Interestingly, despite the increased mortality rates identified in numerous studies involving patients with G5-stage CKD, the risk of death continues to rise among those requiring renal replacement therapy [6]. When comparing patients treated with peritoneal dialysis (PD) to those undergoing hemodialysis (HD), it becomes evident that hemodialysis carries a higher risk of cardiovascular death [7]. The situation is further complicated by the fact that based on the current drug treatment (e.g., lipid-lowering, blood-pressure-regulating medications, SGLT2 inhibitors, GLP1 analogs, among others) and prevention procedures recommended by KDIGO 1a evidence, the risk of cardiovascular death in patients with G4-stage CKD is expected to decline. This underscores the quantitative disadvantage of HD patients [8], at least for the first 2.5 years on dialysis [9,10]. The higher cardiovascular mortality associated with G5-stage CKD raises important questions about the underlying factors that contribute to this increased risk in patients undergoing hemodialysis. In addition to the well-known risk factors, chronic inflammation has also been proven to play a role in cardiovascular mortality, even in the early stages of CKD [11]. HD treatment has saved millions of lives over the past fifty years. However, due to the use of artificial materials in HD devices, key components of the immune system, including those of the complement system, are activated, leading to chronic inflammation [12]. Moreover, indwelling vascular dialysis catheters are the other well-known culprits contributing to chronic inflammation [13,14,15]. Several publications have already highlighted that HD patients often develop chronic inflammation, which is linked to a higher risk of cardiovascular complications and death. These studies have demonstrated increased levels of inflammatory mediators and cytokines in the blood of HD patients [16,17,18,19]. Our hypothesis for this review paper was that immune system activity that is elevated due to artificial materials and direct exposure to the bloodstream during HD may mediate the elevated risk of cardiovascular death when compared with patients receiving PD. As such, we will outline the potential factors that may contribute to the increased immune system activity experienced in HD.

## 2. Materials and Methods

We searched medical databases, including PubMed and Google Scholar, using keywords such as “chronic uremic inflammation”, “hemodialysis treatment-related inflammation”, “cardiovascular mortality in HD patients”, “mortality between HD and PD”, “complement system activation in hemodialysis”, and “hemodialysis procedure-related reactions”. This search aimed to identify the relevant scientific literature on these topics. Furthermore, we attempted to determine whether we could identify cited publications based on the correlations between the given search terms, e.g., “hemodialysis treatment-related inflammation and cardiovascular mortality in HD patients”. The search process was performed systematically, and the identified articles were carefully evaluated and reviewed for their relevance and quality. We followed the guidelines for the literature search and manuscript preparation [20,21]. The present study established inclusion criteria for all articles published by 1 May 2025, including systematic reviews and studies involving humans or animals. We utilized pertinent publications that aligned with the search terms outlined in the narrative review, which may provide a foundation to support our hypothesis.

In this narrative review, we focused exclusively on non-traditional risk factors. This is because they are well-known and have been discussed in numerous publications. However, little has been done to explore the relationship between mortality and morbidity between HD and PD patients. We hypothesize that ESRD patients are at the same cardiovascular risk but that this risk is higher in HD patients. The increased cardiovascular risk in HD patients may be due to an increased inflammatory response, which is associated with HD treatment.

## 3. Results

Based on the above, in our narrative review statement, we aimed to highlight that cardiovascular morbidity and mortality are well-known to be high in patients with end-stage renal disease. Due to chronic inflammation and potentially hemodialysis-related pathological processes, mortality in hemodialysis patients exceeds that of peritoneal dialysis patients. Therefore, we also aimed to find out whether there are any publications in the literature discussing this difference in mortality. The search results obtained for the keywords are summarized in Table 1.

Most publications focus on the results and conclusions of human observational studies. A significant number of publications have examined cardiovascular mortality among HD patients. However, most of them discuss traditional risk factors primarily. They also analyze acute or chronic inflammatory processes related to HD treatment from the perspective of various bloodstream and other infections. Surprisingly, only a few publications have explored the connection between hemodialysis treatment-related inflammation and cardiovascular mortality in HD patients. Therefore, we believe it is important to emphasize the chronic inflammatory processes in end-stage renal disease discussed in our manuscript, as understanding these processes could lead to the development of future therapeutic options.

## 4. Chronic Uremic Inflammation

As kidney function gradually declines, there is a corresponding increase in various uremic waste products. These uremic toxins can impact multiple organ systems and physiological functions. It is essential to distinguish between “urea”, one of the least harmful substances that accumulates during uremia, and the broader category of “uremic toxins”, which encompasses a wide range of substances, including urea [22,23].

Uremic toxins are categorized into three classes based on their molecular weight and interactions: 1. small, water-soluble uremic toxins, such as urea and creatinine, 2. protein-bound uremic toxins, such as indoxyl–sulfate and p-cresyl sulfate, and 3. middle molecular weight peptides or proteins, such as beta2-microglobulin, parathyroid hormone (PTH), interleukin-6 (IL-6), and immunoglobulin light chains. Table 2 summarizes the most important uremic toxins generated in CKD.

Other mediators of uremic toxicity, such as lipopolysaccharides (endotoxins) and advanced glycation end products (AGEs), do not belong to these well-established categories. The accumulation of AGEs is hazardous in the populations affected by CKD and dialysis. AGEs build up in tissues through covalent binding via a non-enzymatic reaction between reducing sugars and the free amino groups of proteins, lipids, and nucleic acids. While diabetes is typically recognized as the primary disease driving AGE formation, CKD is also closely linked to AGE accumulation, which is attributed to reduced renal excretion, increased oxidative stress, and higher levels of precursor compounds [24]. Multiple studies suggest that factors associated with CKD, such as systemic chronic low-grade inflammation, heightened oxidative stress, and uremic toxins [25], contribute to the acceleration of atherosclerosis in CKD patients. Moreover, uremic toxins chronically activate toll-like receptors (TLRs), an integral part of innate immunity, and elicit the production of pro-inflammatory danger-associated molecular patterns (DAMPs), such as high-mobility group box 1 (HMGB1). This process exacerbates the inflammatory environment. This detrimental cycle is further perpetuated by activating additional DAMP receptors beyond TLRs, including NLR-inflammasome-activated caspase-1 and various other pro-inflammatory cytokines. These factors elevate the levels of uremic toxins and inhibit the function of T helper cells (CD4+ cells) [26]. Even early in the course of the disease, systemic chronic low-grade inflammation and elevated oxidative stress can be detected in CKD patients, as evidenced by high levels of circulating inflammatory proteins (including C-reactive protein [CRP] and IL-6) as biomarkers of oxidative stress [27]. Oxidative stress is a significant factor contributing to the progression and complications of CKD and ESRD. Numerous pathological processes associated with oxidative stress have been identified and investigated in patients with CKD and ESRD within a uremic environment. A marked decrease in antioxidant enzyme activity and elevated lipid peroxidation play a crucial role in this phenomenon. Figure 1 summarizes the main pathway involved in chronic inflammation and organ damage related to CKD. As explained earlier, the ongoing inflammatory state and oxidative stress caused by the worsening of kidney damage eventually lead to further damage to the target organs.

In addition, uremic-serum-induced oxidative stress in endothelial cells, the damage of which promotes vascular calcification [28], is evidenced by the increased generation of reactive oxygen species (ROS) [29] through a RAGE-NF-κB-dependent pathway. This signaling pathway, along with glutathione S-transferase μ1, which is the product of GSTM-1, a downstream gene in the aryl-hydrocarbon-receptor (AhR) signaling cascade, contributes to the uremic-serum-mediated up-regulation of adhesion molecules in endothelial cells [30]. Among the inflammatory mediators dysregulated in CKD, increased levels of vascular cell adhesion protein 1 (VCAM-1) and the pro-inflammatory cytokine monocyte chemoattractant protein-1 (MCP-1) in uremic endothelial cells result in increased monocyte adhesion and, consequently, increased inflammation [25,31]. As previously discussed, oxidative stress resulting from uremic toxins, which is associated with endothelial injury and vascular calcification, is evident in patients with HD and PD. An important factor may also be the significant periodic fluid overload documented based on elevated CRP in HD patients who were volume overloaded when volume status was measured through bioimpedance. Fluid-overloaded HD patients are known to have significantly higher CRP, lower albumin, and a higher neutrophil-to-lymphocyte ratio [32]. Nonetheless, these phenomena do not account for all of the observed differences in mortality rates between individuals undergoing HD and PD.

## 5. Cardiovascular Mortality in HD Patients

The conventional risk factors for cardiovascular diseases are diabetes, atherosclerosis, hypertension, hyperlipidemia, and smoking. These cardiovascular risk factors are well-known and discussed in several publications [33,34,35,36]. Both diabetic and nondiabetic patients with coronary artery disease who underwent PD exhibited a significantly elevated risk of mortality compared to those receiving HD. In contrast, nondiabetic patients without coronary artery disease demonstrated comparable survival rates regardless of whether they were treated with PD or HD [37]. Advanced vascular calcification is a critical factor in ESRD progression associated with metabolic bone disorder (CKD-MBD). CKD-MBD is a complex clinical syndrome encompassing disorders of phosphate, PTH, calcium, vitamin D, and fibroblast growth factor-23 (FGF23) metabolism [38]. A particularly severe manifestation of this condition is uremic calcific arteriolopathy or calciphylaxis [39,40]. Figure 2 highlights the additional effects of CKD, especially cardiovascular issues. As kidney function decreases, different problems linked to impaired kidney activity, such as the activation of the renin–angiotensin–aldosterone system (RAAS), hypervolemia, and renal anemia, become apparent. The kidneys’ reduced capacity to remove waste causes the buildup of uremic toxins, along with oxidative stress and ongoing inflammation.

All of these pathological processes are accompanied by chronic inflammatory responses associated with the uremic environment. This interplay further enhances the morbidity and mortality associated with traditional cardiovascular risk factors [11,41]. Well-known cardiovascular risk factors affect both HD and PD patients. However, mortality in HD patients is higher compared to PD patients. Although the difference in mortality is evident in the first years, this difference is eventually no longer visible due to peritoneal fibrosis with prolonged duration of PD treatment. These differences indicate the role of the direct immunological effects of HD treatment. Unlike during hemodialysis, PD solution does not come into direct contact with the patient’s blood [42]. To recite, PD offers a unique modality to separate the dialysis procedure from the bloodstream and a disproportionate removal gradient of uremic substances from the abdominal compartment [43]. However, peritoneal fibrosis that develops over time eventually leads to ultrafiltration failure and reduced efficiency of the modality, which plays a role in the increased cardiovascular mortality in PD patients. This may explain why the mortality difference between HD and PD disappears after a few years [44]. By triggering a systemic stress response, HD inherently exacerbates patient morbidity and mortality. This condition arises from factors such as hemodynamic management strategies (including weight loss and ultrafiltration), treatment protocols, solute transport dynamics, electrolyte shifts, and the interactions between the circulating blood and extracorporeal circuit ECC [45].

## 6. Hemodialysis Treatment-Related Inflammation

The prevalence of patients with ESRD requiring HD treatment is rising globally. A variety of medications are prescribed to treat CKD-related complications in affected individuals [46]. Since the beginning of renal replacement therapy with maintenance HD, a variety of dialysis-related adverse reactions have been registered. These reactions are observable and of immediate significant clinical importance for practicing clinicians. Based on their underlying mechanisms, they are grouped into two main categories: type A (hypersensitivity reactions, HSRs) and type B (nonspecific reactions). Studies suggest that the incidence of these reactions is approximately 4 in 100,000 cases of HD for type A reactions, while type B reactions occur in about 3 to 5 cases per 100 treatments [47]. According to this idea, type A reactions usually start within minutes of initiating HD treatment and are predominantly mediated by immunoglobulin E (IgE) [48,49]. The resulting HSR is often associated with severe clinical symptoms and, in all cases, requires the immediate discontinuation of HD treatment [29]; otherwise, the affected patient may develop serious complications. Mild clinical symptoms include itching, a burning sensation in the fistula arm, urticaria, flushing, cough, dizziness, epigastric discomfort, limb cramps, diarrhea, headache, nausea, chest pain, and fever. More severe reactions to dialysis can manifest as anaphylactic shock, dyspnea, and hypotension, which, if not treated promptly, could lead to cardiac arrest or even death [50,51]. Recent data suggest that substantial portions of acute allergic reactions, termed as anaphylactoid, pseudoallergic, or infusion reactions, are non-IgE-mediated [52]. In contrast, according to the HD-induced HSR literature, type B reactions typically present with a milder clinical course. Their underlying cause is the activation of the complement system, which leads to the associated symptoms. These reactions often occur within the first 30 min of HD treatment. The characteristic symptoms of type B dialysis reactions include chest and back pain, dyspnea, nausea, vomiting, and hypotension, which generally do not necessitate the immediate discontinuation of HD treatment [53]. Several triggering mechanisms have been postulated, including, in addition to the activation of the complement system, the accumulation of white blood cells in the lungs, sensitization induced by ethylene oxide used to sterilize the extracorporeal system, a reaction between angiotensin-converting enzyme inhibitors and the AN69 membrane, or even contamination of the dialysis fluid [54,55,56]. Decades ago, reactions associated with HD treatment were most often caused by the ethylene oxide sterilizer or the use of less biocompatible dialysis membranes. Reactions during HD treatment, particularly those that manifest in a latent form and occur repeatedly, may further exacerbate the chronic inflammation sustained by the uremic environment. Ultimately, this may contribute to the elevated mortality rates observed in patients undergoing HD. In contrast, PD treatment is associated with fewer immunological reactions, but a distinct separate issue is the biocompatibility of PD fluids and PD-induced changes in parietal and visceral serous surfaces in the abdominal cavity. Over time, the peritoneal solute transport rate tends to increase in a subset of PD patients, leading to ultrafiltration failure and decreased removal of uremic waste products. Prolonged exposure of the peritoneum to hypertonic glucose solutions may result in morphological damage, potentially contributing to alterations in peritoneal function [57,58]. The types of PD solutions in use influence and modulate the inflammatory response in PD. Currently, a variety of PD fluids are available on the market, including (a) conventional solutions, (b) neutral pH solutions containing low concentrations of glucose degradation products, (c) solutions with icodextrin, and (d) fluids enriched with taurine [44]. Repeated episodes of infectious peritonitis may be another potential cause of chronic inflammation [59].

## 7. Complement System Activation in Hemodialysis

Recognition of the relationship between complement (C) system pathways and HD therapy originated in the early days of maintenance dialysis. In the late 1960s and early 1970s, Craddock et al. noted that acute pulmonary dysfunction may result from complement-mediated leukostasis when using cellulose-based membranes, particularly cuprophan, which was widely used to meet the increasing demand for HD [60,61]. Significantly higher levels of C3a and C5a were observed on the venous side during HD treatment, indicating activation of the complement system through the alternative pathway. This activation occurs due to the dialyzer used in the HD. In the first 10 to 15 min of HD, increased C3a levels were documented, indicating C3 activation. This was followed by elevated C5a and C5b levels, with soluble C5b-9 levels rising by up to 70%, and the plasma C3d/C3 ratio increasing during the session [62]. Additionally, in cases where anaphylactic reactions occurred during HD treatment, significantly elevated levels of C3a and C5a were measured [63]. Figure 3 illustrates the process of complement activation during HD. The hemodialysis membrane can trigger the complement system, starting with ficolin-2 binding to the dialyzer, which activates the lectin pathway. Meanwhile, properdin or C3b activates the alternative pathway. This activation increases C3 receptors on leukocytes, leading to their accumulation on the dialyzer’s surface and resulting in heightened inflammation and a risk of thrombosis.

Recent observations of anaphylatoxin (C5a, C3a) formation associated with various dialyzer membranes have drawn increased attention to HD-related bio-incompatibility issues. As a result, the use of cellulose-based HD membranes has largely declined, with a significant shift in favor of synthetic polymer membranes, particularly those constructed from polysulfone [64,65,66]. Polysulfone membranes can also activate the complement system, leading to mast cell degranulation and the subsequent release of various mediators. When the complement system is activated, C3a, C5a, and other anaphylatoxins are produced, contributing to the onset of systemic symptoms [67]. The composition and structural properties of the dialyzer membrane significantly influence the activation of the complement system and the subsequent reactions that occur during HD treatment [68]. However, the pathomechanism of reactions taking place during the use of polysulfone membranes is very different from reactions observed with cellulose membranes described earlier [69]. Polysulfone membranes can induce type B reactions, which are characterized by clinically nonspecific symptoms [54]. Type B reactions have somewhat similar characteristics to the so-called complement activation-related pseudoallergy (CARPA) [52], with clear similarities but also clear-cut discrepancies between the two.

CARPA is a drug-induced immediate HSR, the onset of which occurs within minutes. Its symptoms fit in Coombs and Gell’s type I category of HSRs (of the four traditional groups from I to IV) [70], but they are not actually initiated or mediated by pre-existing IgE antibodies. Instead, its common triggering mechanism is the activation of the complement system. Pseudoallergic reactions represent a high percentage of all immune-mediated immediate HSRs [52] induced by a variety of therapeutic and diagnostic agents, which belong to nanoparticles (NPs) [71]. In some cases, e.g., in liposome-induced infusion reactions, in addition to complement activation, the involvement of IgG and IgM is also demonstrated [72]. Notably, sensitivity to CARPA is particularly pronounced in pigs. Therefore, the porcine CARPA model is the benchmark for testing HSRs caused by various NPs, the symptoms of which are widely documented. [71,72,73,74,75,76,77,78,79,80,81,82,83]. The severity of porcine CARPA may vary from mild to severe hemodynamic, cardiopulmonary, hematological, or skin reactions. The most severe outcome involves anaphylaxis, which requires cardiopulmonary resuscitation [81]. During the cleavage of C3 and C5, anaphylatoxins C3a and C5a and opsonins C3b and C3dg are released, which trigger HSRs. The released products of the complement cascade subsequently stimulate the release of inflammatory cytokines, including interleukin-1 beta (IL-1β), tumor necrosis factor-alpha (TNF-α), and interleukins 6 and 8 (IL-6 and IL-8). In conjunction with the activation of the complement system, the up-regulation of specific cell surface receptors on leukocytes, such as CD11b/CD18 and CD35, is observed, coupled with the down-regulation of L-selectin [84]. Moreover, in HSRs, oxidative stress induced by white blood cells is associated with dysfunction in endothelial cells. This association supports the hypothesis that the increased release of inflammatory cytokines due to complement system activation during hemodialysis treatment ultimately contributes to increased oxidative stress, resulting in vascular calcification. This phenomenon may account for the higher mortality rates observed in patients undergoing HD [85,86].

## 8. Hemodialysis Procedure-Related Reactions

Membranes play an essential role in activating the immune system. A more recent advancement in understanding the phenomenon of bio-incompatibility involves the recognition of the importance of neutrophil extracellular traps (NETs), known for their participation in NETosis, a significant detrimental process observed in various pathophysiological conditions [87,88]. The induction of NETosis leads to neutrophil degranulation and the production of reactive oxygen species (ROS), primarily generated by nicotinamide adenine dinucleotide phosphate (NADPH) oxidase enzymes. While NETs have beneficial antimicrobial properties, they may also contribute to the pathogenesis of various diseases [89]. ESRD and the HD procedure serve as significant catalysts for forming neutrophil extracellular traps (NETs) [88]. The association between the dialysis process and membrane-induced NETosis highlights another dimension of the complex phenomena related to bio-incompatibility [90,91,92]. Considering bio-incompatibility and the formation of NETs, one can hypothesize that the uremic environment—consisting of complement and inflammatory marker uremic toxins, contaminants in the dialysis fluid, such as lipopolysaccharide (LPS), along with the artificial surfaces of the membrane and the ECC—collectively or individually serve as stimuli for NETs’ generation during HD. Moreover, these elements may directly contribute to reactions that induce dialysis-related systemic stress conditions It is highly plausible that NETs remain active during the interdialytic period and may contribute to mortality risk [91].

In our previous animal study in pigs, acute pulmonary hypertension (PH) occurred following the reinfusion of blood that had been circulated through an ECC of a dialysis machine. This process activated the complement system, resulting in higher C3a and C5a levels and acute hemodynamic changes [93]. Another notable finding in this porcine experimental HD model was that the inflammatory markers, especially thromboxane B2 (TxB2), increased in parallel with pulmonary artery pressures (PAP) after the extracorporeal blood was rinsed back into the pigs at the end of the HD treatment. We could observe this rise in TxB2 and PAP levels when a cuprophan dialyzer was used, and a similar reaction was elicited even with a stand-alone extracorporeal circulation setup without a dialyzer incorporated [94,95]. Based on the results of observational studies, PH occurs in 10-70% of chronic and end-stage renal disease patients, which is an independent risk factor for increased mortality [96,97,98]. According to our hypothesis, the increase in TxB2 observed in animal experiments may be crucial in developing human PH in ESRD on HD. The extracorporeal circulation plays a pivotal role in the inflammatory cascade associated with HD treatment, similarly to the immunological reactions observed during cardiopulmonary bypass surgeries. However, the former represents repeated insults, with dialysis sessions occurring thrice weekly [99]. This process is initiated by systemic inflammatory response syndrome (SIRS), which involves significant contributions from both pro-inflammatory and anti-inflammatory cytokines, the complement system, and neutrophils. The alternative pathway is activated when the patient’s blood comes into contact with the extracorporeal circuitry representing “foreign material”; consequently, C3a and C5a are formed. Meanwhile, the classic pathway is triggered by heparin reversal with protamine, leading to increased C4a and subsequent C3a formation [100,101,102]. The consequence of the activation of the complement system is the release of inflammatory mediators [103,104,105,106].

## 9. Discussion

To summarize, CKD is a progressive loss-of-function condition resulting in wide repercussions for all human body functions. The relationship between CKD and CVD involves several interconnected risk factors [107,108,109], including hypertension, which both causes and exacerbates CKD and further exacerbates cardiovascular risk [110,111]. Diabetes is the primary global cause of CKD [112,113,114]. Diabetic patients often display abnormal lipid profiles, including elevated triglycerides and low HDL cholesterol, promoting plaque formation and atherosclerosis, preceding even the period of CKD [115,116,117,118,119]. CKD disrupts calcium and phosphate balance, leading to vascular calcification, which stiffens blood vessels and raises cardiovascular risk [120]. Additionally, anemia from reduced erythropoietin production in CKD further strains the heart, potentially contributing to left ventricular hypertrophy and heart failure [121,122,123,124,125]. Mortality rates among dialysis patients are higher in younger age groups, primarily due to cardiovascular (40%) and infectious causes (10%). High cardiovascular mortality may be linked to shared risk factors like chronic inflammation, changes in extracellular volume, dystrophic vascular calcification, and altered cardiovascular dynamics during dialysis [126]. Cardiovascular disease impacts more than two-thirds of individuals undergoing HD, serving as a significant contributor to morbidity and accounting for nearly 50% of mortality in this population [127]. An interesting finding is that there were no significant differences in the risk of death from cardiovascular issues or infections when analyzed by sex. This emphasizes the importance of recognizing both traditional and non-traditional risk factors relevant to each gender [128]. Most publications mainly highlight the increased risk of cardiovascular death linked to traditional risk factors in patients on HD [85,129,130]. This is why we believe it is essential to address non-traditional risk factors that contribute to the increase of cardiovascular mortality.

As reviewed before, patients with CKD undergoing HD display an increased inflammatory response. According to our hypothesis, the heightened inflammatory response observed in HD patients acts as a non-traditional risk factor contributing to increased cardiovascular mortality. This phenomenon is likely due to the direct exposure of blood to the bio-incompatible conditions of HD, a phenomenon that does not present in PD. It may also stem from the variable nature of volume shifts and the different removal of uremic toxins associated with HD in contrast to PD [131,132,133]. Numerous studies have demonstrated that patients undergoing HD treatment exhibit an elevated inflammatory state, which is associated with an increased risk of cardiovascular complications and heightened mortality rates [134]. Chronic inflammation mainly results from various infections encountered during HD treatment. Vascular access points are the main entry sites for bloodstream infections. These infections, along with the resulting inflammatory responses, are significant non-traditional risk factors that contribute to the increased cardiovascular mortality linked to HD [135,136,137]. In addition to the traditional and non-traditional risk factors previously discussed, our hypothesis suggests that it is important to consider the role of dialyzers and devices used during hemodialysis treatment. As artificial materials, these can contribute to both acute and chronic inflammatory responses. While manufacturers aim to develop biocompatible materials, adverse reactions can still occur [138]. As discussed previously, several factors can elicit acute anaphylactoid reactions during hemodialysis. Specific dialysis membranes, particularly those from older generations, can provoke such reactions. Membranes composed of materials like cellulose or modified cellulose have been linked to a higher incidence of these reactions due to their ability to activate complement pathways, resulting in mast cell degranulation [49,63,139]. Due to its large surface area, the dialysis membrane is the primary cause of HD-related reactions, as it is the point of the most significant interaction between the patients’ blood and foreign materials. This understanding has led to efforts aimed at creating more biocompatible membranes. However, even with the widespread use of modern polysulfone membranes, we still observe both acute and likely chronic inflammatory responses [53,140,141]. In light of the undesirable side effects associated with polysulfone membranes, further efforts have been made to improve biocompatibility, including integrating various antioxidant substances on the membrane’s surface, such as tocopherol. This approach enhances biocompatibility and mitigates the chronic inflammatory response induced by the dialysis membrane, potentially reducing the risk of cardiovascular mortality [142,143,144,145,146,147]. Elevated complement activity is often observed in the context of acute and inflammatory processes linked to HD treatment. Numerous clinical studies have verified the activation of the complement system during HD, leading to increased levels of C5b-9. This elevation of C5b-9 is closely associated with higher cardiovascular mortality [62,148,149].

In our narrative review, we aimed to explore the reasons for the higher rates of cardiovascular mortality. Our narrative review suggests that HD treatment may contribute to an increased inflammatory response, possibly explaining the rise in cardiovascular deaths. Several studies show that patients on HD have noticeably higher mortality rates compared to the general population [150]. Similarly, patients receiving PD also experience increased mortality rates relative to the general population. However, when these two modalities are compared head to head, PD may have a marginally lower mortality rate during the initial years of treatment [42,151,152]. Supporting the hypotheses, HD treatment is linked to higher overall all-cause mortality compared to PD according to a recently published systematic review [153]. The initial mortality rates associated with PD are more favorable; however, they eventually align with those of patients undergoing HD. This shift can be attributed to complications arising from prolonged PD treatment, which can result in ultrafiltration failure and inadequate dialysis with a progressive loss of residual kidney function and the failure of endogenous functions of the peritoneum. Without these complications, the mortality risk would not be equivalent between PD and HD patients [154,155]. Previous analyses have elucidated the significant correlation between chronic inflammation, cardiovascular disease, and heightened mortality rates. Numerous studies indicate that patients undergoing HD exhibit an augmented inflammatory response, which is closely associated with the treatment modalities employed in HD [156,157,158,159]. Among the factors contributing to bio-incompatibility reactions in HD, the dialysis membrane itself stands out as a significant trigger for complement activation-related inflammatory responses. Thus, the heightened and complex inflammatory response specifically associated with HD may account for the increased mortality rates observed in HD-treated patients. This phenomenon could explain the notably lower mortality among patients undergoing PD during the initial treatment period, as PD does not elicit the same inflammatory response as HD [160,161,162,163]. These responses invoke further cell senescence, inflammation, NETosis, and CV changes. In addition, our own results suggest that the extracorporeal setup activating reinfused blood components could contribute to elevated pulmonary pressures [94]. Improving the biocompatibility of dialysis materials is essential for mitigating systemic inflammation and enhancing the clinical outcomes of patients undergoing dialysis. In terms of immediate future technical innovation, data analysis with artificial intelligence (AI) technology may bring immediate results in this field. Despite several limitations and while not aiming to replace human judgment, AI can assist in everyday medical decision making. AI tools can enhance existing risk assessment methods, facilitating the early identification of high-risk patients who could benefit from personalized interventions and intensified monitoring. These measures may ultimately help reduce mortality, particularly among patients undergoing frail HD treatment [164].

## 10. Conclusions

In addition to traditional risk factors, the heightened cardiovascular morbidity and mortality in hemodialysis patients may be attributed to the increased inflammatory response directly triggered by hemodialysis treatment. During hemodialysis treatment, published results highlight the potential activation of the complement system caused by artificial materials, which occurs despite improved biocompatibility. Additionally, immune cells sequestered in the extracorporeal system, when returned to the body at the end of the treatment, trigger a detectable inflammatory response. Recent research and results also highlight the role of NETosis in patients undergoing hemodialysis. The lower mortality rates observed during the initial years of PD treatment dissipate with the increasing duration of PD. Hypothetically, if the effectiveness of PD treatment remained constant over time, mortality rates would be significantly lower, alongside reduced inflammatory activation. The heightened inflammatory response seen during hemodialysis may represent an attractive future target aimed at decreasing cardiovascular morbidity and mortality in patients undergoing hemodialysis. The therapeutic inquiry posits whether the development and utilization of immunomodulatory agents in conjunction with hemodialysis may effectively diminish the inflammatory response and, as a result, mitigate cardiovascular mortality.

## 11. Limitations and Future Directions

Our narrative review has several limitations that should be noted. First of all, due to the extent of this topic, we focused only on those unconventional risk factors that may play a role in the increased cardiovascular mortality of HD patients and show a relationship with HD treatment. Furthermore, the quantitative relationship between the non-conventional risk factors and outcomes remains poorly understood. Accordingly, future research must investigate the associations between non-conventional risk factors, increased cardiovascular mortality, and the heightened inflammatory response observed with HD treatment in large cohorts of patients.

## Figures and Tables

**Figure 1 toxins-17-00345-f001:**
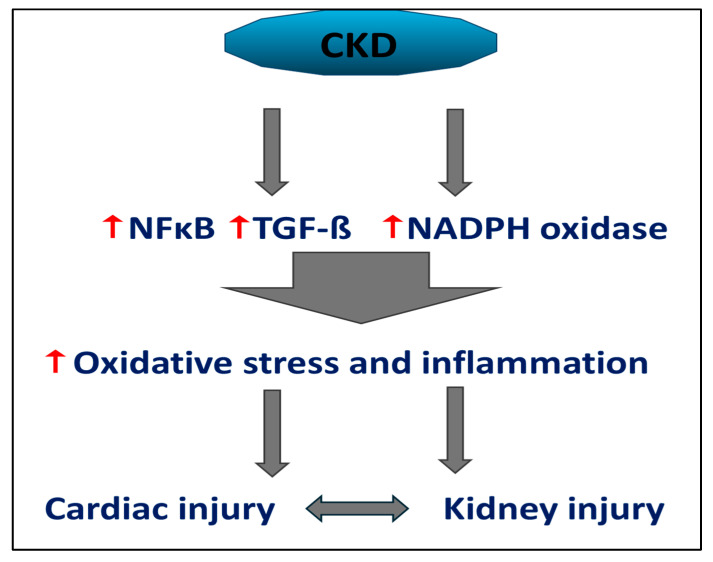
In chronic kidney disease, the release of reactive oxygen radicals and increased inflammatory mediators lead to oxidative stress and persistent inflammation. This results in damage to the heart and kidneys, creating a harmful cycle. (Abbreviations: CKD: chronic kidney disease; NADPH: nicotinamide adenine dinucleotide phosphate; NFκB: Nuclear factor kappa-light-chain-enhancer of activated B cells; TGF-β: transforming growth factor beta).

**Figure 2 toxins-17-00345-f002:**
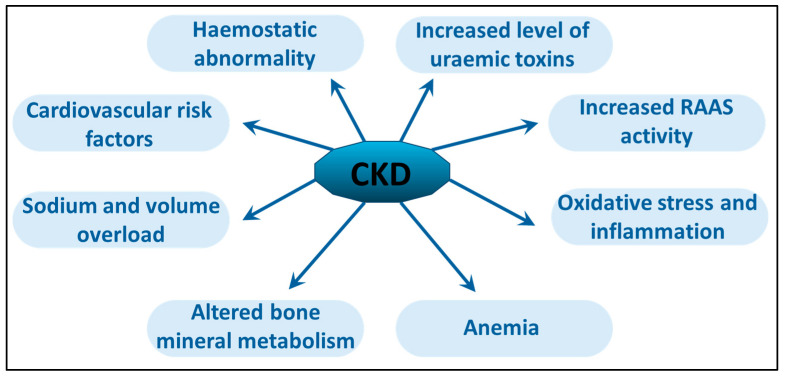
Chronic kidney disease causes damage to multiple organs and leads to various complications. As kidney function declines, issues arise from reduced nephron numbers and fibrosis, including activation of the RAAS system, hypervolemia, and renal anemia. The kidneys’ diminished ability to excrete waste is linked to the accumulation of uremic toxins, oxidative stress, and chronic inflammation. (Abbreviations: CKD: chronic kidney disease; RAAS: renin–angiotensin–aldosterone system).

**Figure 3 toxins-17-00345-f003:**
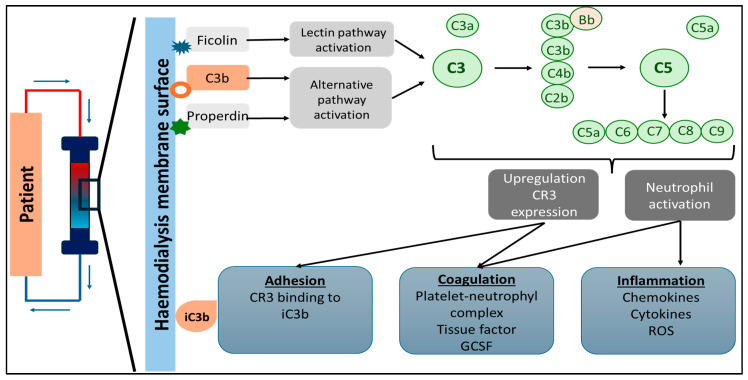
The hemodialysis membrane can activate the complement system, triggering the release of mediators from mast cells. This process starts with ficolin-2 binding to the dialyzer, activating the lectin pathway, while properdin and/or C3b initiate the alternative pathway. This activation increases C3 receptors on leukocytes, leading to their sequestration on the dialyzer’s surface, resulting in heightened inflammation and a potential risk of thrombosis. (Abbreviations: C: complement; GCSF: granulocyte colony-stimulating factor; ROS: reactive oxygen species).

**Table 1 toxins-17-00345-t001:** The search results for the keywords “chronic uremic inflammation”, “hemodialysis treatment-related inflammation”, “cardiovascular mortality in HD patients”, “mortality between HD and PD”, “complement system activation in hemodialysis”, and “procedure-related reactions”.

Search Keywords	Number of Publications Until 1 May 2025	First Publication in This Field
chronic uremic inflammation	1284	1966
hemodialysis treatment- related inflammation	2845	1969
cardiovascular mortality in HD patients	2389	1968
hemodialysis treatment- related inflammation and cardiovascular mortality in HD patients	100	1998
mortality between HD and PD	4907	1968
complement system activation in hemodialysis	423	1964
hemodialysis procedure- related reactions	1305	1964

**Table 2 toxins-17-00345-t002:** Classification of uremic toxins produced in chronic kidney disease based on their molecular weight.

Small Molecular Weight (<500 Da)	Middle Molecular Weight (≥500 Da)	High Molecular Weight (Mostly < 500 Da)
Urea	PTH	Indoxyl sulfate
Creatinine	β2-microglobulin	Indole acetic acid
Uric acid	Endothelin	*p*-cresylsulfate
ADMA	FGF23	Phenylacetic acid
Carbamylated compounds	Ghrelin	Kynurenines
SDMA	Immunoglobulin light chains	AGEs
TMAO	IL-6, IL-8, IL-18	Homocysteine
	Lipids and lipoproteins	
	Neuropeptide Y	
	ANP	
	Retinol binding protein	
	TNF-α	

Abbreviations: ADMA: asymmetric dimethyl-arginine; IL: interleukin; SDMA: symmetric dimethyl-arginine; TMAO: trimethyl-amine-*N*-oxide; ANP: atrial natriuretic peptide; FGF23: fibroblast growth factor 23; PTH: parathyroid hormone; TNF-α: tumor necrosis factor-α, AGEs: advanced glycation end products.

## Data Availability

No new data were created or analyzed in this study.

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
