# Peer review of "Increased Cardiovascular Mortality in Hemodialysis: The Role of Chronic Inflammation, Complement Activation, and Non-Biocompatibility"

_toxins, 2025, doi:10.3390/toxins17070345_

Round 1
Reviewer 1 Report
Comments and Suggestions for Authors
Dear Authors,
The article addresses a relevant topic with potential clinical implications. However, its methodological quality, the wording in English, and the presentation of the data require substantial revisions. The article should be reformulated with a more systematic review methodology, improve the language and enrich the discussion.
Major comments
Tthe introduction provides a good context but there are conceptual repetitions. It is recommended to reconvert the content.
Methodology: The literature search is inadequately described. Statements such as “PubMed and Google Scholar were searched” are insufficient and should be better justified. No clear criteria for inclusion or exclusion of selected studies are provided. The search strategy lacks a structured approach. At a minimum, authors should describe their search strategy (databases, dates, search terms, filters), selection criteria, and critical evaluation approach, including within a narrative framework.
Many sections repeat the same concepts several times. Figures (especially Figures 2-4) require clearer legends and integration into the text. Suggestion: Reorganize the content to group related mechanisms (e.g., all complement-related sections) and avoid splitting discussions into several chapters unnecessarily.
Providing more quantitative data (e.g., effect sizes, incidence rates) would strengthen the manuscript.
The discussion should better differentiate between established knowledge and emerging hypotheses.
Minor comments
The use of abbreviations is excessive in some places and inconsistent in definition (e.g., HD, PD, ECC).
More recent references or studies of high methodological quality could be included. There is an overload of references prior to 2010.
The article deals with a relevant topic but needs to be modified.
Comments on the Quality of English LanguageThe manuscript contains frequent grammatical issues, awkward syntax, and imprecise vocabulary. A professional English editing service is strongly recommended to improve clarity, readability, and academic tone.
Author Response
The article addresses a relevant topic with potential clinical implications. However, its methodological quality, the wording in English, and the presentation of the data require substantial revisions. The article should be reformulated with a more systematic review methodology, improve the language and enrich the discussion.
Answer: Thank you for your comment. Thank you, we also believe that the topic you raised is very important and may be of interest to readers. We have tried to improve our manuscript to the best of our ability based on the suggestions. After correcting our manuscript, we requested an English language proofreading.
Major comments
The introduction provides a good context, but there are conceptual repetitions. It is recommended to reconvert the content.
Answer: Thank you for your comment. We made some corrections in the introduction. The main message and hypothesis of our manuscript is that although mortality rates should be the same in all ESKD patients, regardless of renal replacement modality, mortality rates are still higher in patients treated with HD. Therefore, we have detailed mortality and risk factors.
Methodology: The literature search is inadequately described. Statements such as “PubMed and Google Scholar were searched” are insufficient and should be better justified. No clear criteria for inclusion or exclusion of selected studies are provided. The search strategy lacks a structured approach. At a minimum, authors should describe their search strategy (databases, dates, search terms, filters), selection criteria, and critical evaluation approach, including within a narrative framework.
Answer: Thank you for your comment. Based on the reviewer suggestions, we restructured this section and added an additional “results” section to our manuscript.
Many sections repeat the same concepts several times. Figures (especially Figures 2-4) require clearer legends and integration into the text. Suggestion: Reorganize the content to group related mechanisms (e.g., all complement-related sections) and avoid splitting discussions into several chapters unnecessarily.
Answer: Thank you for your comment. We believe that the chapters "Complement System Activation in Hemodialysis" and "Hemodialysis Procedure-Related Reactions" should be discussed separately. The activation of the complement system during HD treatment is discussed in detail in numerous publications; therefore, we consider it justified to briefly discuss it separately. Research on reactions related to HD treatment has begun in recent years. This chapter is crucial, as it fundamentally determines the outcome of patients treated with HD and, of course, the development of biocompatible materials. A more detailed description of the figures is a completely legitimate suggestion, and we have corrected these in the manuscript.
Providing more quantitative data (e.g., effect sizes, incidence rates) would strengthen the manuscript.
Answer: Thank you for your comment. We added an additional “results” section to our manuscript
The discussion should better differentiate between established knowledge and emerging hypotheses.
Answer: Thank you for your comment. We added additional explanations to the discussion section.
Minor comments
The use of abbreviations is excessive in some places and inconsistent in definition (e.g., HD, PD, ECC).
Answer: Thank you for your comment. We reduced the number of abbreviations.
More recent references or studies of high methodological quality could be included. There is an overload of references prior to 2010.
Answer: Thank you for your comment. We have added additional references to our manuscript. However, it is true that the description and discussion of reactions related to HD treatment in particular was more typical of earlier years, and relatively fewer publications on this topic have been published in recent years.
The article deals with a relevant topic but needs to be modified.
Answer: Thank you for your comment. We thank you for the criticism, and we have tried to correct the errors to the best of our knowledge.
Reviewer 2 Report
Comments and Suggestions for Authors
Dear Authors,
Thank you for the opportunity to review your manuscript again.
I appreciate the improvements you’ve made; however, several issues still need to be addressed to enhance the clarity and quality of the paper:
- Lines 33–34: Please revise the phrase"In practical terms," for improved clarity and academic tone.
- Lines 37–38: The abbreviationCKD (chronic kidney disease) appears twice. Please revise this to avoid redundancy.
- Lines 35 and 50: The phrase"among others" is used in both lines. Consider rephrasing it to improve variation and expression.
- Line 63: There is a typographical error —"may may." Please correct this repetition.
- Figure 1: This figure is placed in the introduction, but it would be more appropriate in the Results section. Alternatively, you may consider removing it, as it presents well-known information.
- Line 73: Please correct the use of the apostrophe (’), which appears to be incorrect.
- Lines 81–82: The sentence,"Relevant publications were identified via a two-pronged approach..." needs further explanation for clarity. Consider elaborating briefly on how the backward and forward search methods were implemented.
- Introduction vs. Methods: The introduction focuses heavily on chronic kidney disease (CKD), while the Methods section discusses end-stage kidney disease (ESKD). Please ensure consistency, or justify this shift clearly in the introduction.
- Lines 92–98: There are too many quotation marks. Simplify the language and avoid redundancy to improve readability.
Additionally, I believe the following article may complement your research well and could be worth referencing:
https://www.mdpi.com/2076-3417/15/10/5776
Author Response
I appreciate the improvements you’ve made; however, several issues still need to be addressed to enhance the clarity and quality of the paper:
Answer: Thank you for your comment.We also consider the topic raised in our manuscript important, because to the best of our knowledge, no such summary has been published. We thank you for the criticism, and we have tried to correct the errors to the best of our knowledge.
Lines 33–34: Please revise the phrase"In practical terms," for improved clarity and academic tone.
Answer: Thank you for your comment. I rewrote this sentence.
Lines 37–38: The abbreviationCKD (chronic kidney disease) appears twice. Please revise this to avoid redundancy.
Answer: Thank you for your comment. I removed the second abbreviation resolution from the sentence.
Lines 35 and 50: The phrase"among others" is used in both lines. Consider rephrasing it to improve variation and expression.
Answer: Thank you for your comment. I rewrote this sentence.
Line 63: There is a typographical error —"may may." Please correct this repetition.
Answer: Thank you for your comment. I removed the unnecessary “may”.
Figure 1: This figure is placed in the introduction, but it would be more appropriate in the Results section. Alternatively, you may consider removing it, as it presents well-known information.
Answer: Thank you for your comment. It is true that Figure 1 does not contain any new information; therefore we removed it from the text.
Line 73: Please correct the use of the apostrophe (’), which appears to be incorrect.
Answer: Thank you for your comment. We corrected it.
Lines 81–82: The sentence,"Relevant publications were identified via a two-pronged approach..." needs further explanation for clarity. Consider elaborating briefly on how the backward and forward search methods were implemented.
Answer: Thank you for your comment. I rewrote this sentence for better clarity.
Introduction vs. Methods: The introduction focuses heavily on chronic kidney disease (CKD), while the Methods section discusses end-stage kidney disease (ESKD). Please ensure consistency, or justify this shift clearly in the introduction.
Answer: Thank you for your comment. The concept of our manuscript was to provide an explanation or hypothesis as to why cardiovascular mortality is higher in patients treated with HD. In order to better understand and summarize the pathological processes, we should mention the uremia-related processes in CKD. After that, we discussed the additional pathological processes associated with HD treatment that are not observed in patients treated with PD. It is quite natural that both HD and PD have the same ESKD-related pathological processes. Yet, mortality is higher in HD.
Lines 92–98: There are too many quotation marks. Simplify the language and avoid redundancy to improve readability.
Answer: Thank you for your comment. I rewrote this paragraph. After correcting our manuscript, we requested an English language proofreading.
Additionally, I believe the following article may complement your research well and could be worth referencing:
https://www.mdpi.com/2076-3417/15/10/5776
Answer: Thank you for your comment. Having read the suggested publication, we added it as a reference to our manuscript.
Round 2
Reviewer 1 Report
Comments and Suggestions for Authors
Dear authors,
Version 2 is better organised. Sections are more clearly delimited. Some paragraphs have been reordered to avoid repetitions which in the first version dispersed the main message.
The tables and figures look a little more refined. The legends have been corrected in some cases for clarity. However, they could still benefit from shorter and more explanatory legends that would allow them to be understood without reference to the text.
A more clinically oriented conclusion, translating molecular findings into practical clinical decisions, is still missing.
Author Response
Version 2 is better organised. Sections are more clearly delimited. Some paragraphs have been reordered to avoid repetitions which in the first version dispersed the main message.
Answer: Thank you for your comment. We express our gratitude, in concurrence with our co-authors, for your constructive suggestions that have significantly enhanced the quality of our manuscript.
The tables and figures look a little more refined. The legends have been corrected in some cases for clarity. However, they could still benefit from shorter and more explanatory legends that would allow them to be understood without reference to the text.
Answer: Thank you for your comment. In all cases, the figure legends have been condensed and rephrased to be more concise.
A more clinically oriented conclusion, translating molecular findings into practical clinical decisions, is still missing.
Answer: Thank you for your comment. We agree that the conclusion lacked a practical clinical message for the reader, so we added this.
Reviewer 2 Report
Comments and Suggestions for Authors
In my opinion, the manuscript can be accepted for publication.
Author Response
In my opinion, the manuscript can be accepted for publication.
Answer: Thank you for your comment. We, along with our co-authors, sincerely appreciate your constructive suggestions that have greatly improved the quality of our manuscript.